# Sulphurous Crenotherapy Is Effective at Reducing Pain and Disability in Overweight/Obese Patients Affected by Chronic Low Back Pain from Spine Osteoarthritis

**DOI:** 10.3390/healthcare10091800

**Published:** 2022-09-19

**Authors:** Maria Costantino, Valeria Conti, Graziamaria Corbi, Irene Ciancarelli, Giovanni Morone, Amelia Filippelli

**Affiliations:** 1Department of Medicine, Surgery and Dentistry “Scuola Medica Salernitana”, University of Salerno, 84081 Baronissi, Italy; 2Association Non-Profit F.I.R.S.Thermae (Interdisciplinary Training, Researches and Spa Sciences) Naples, 80078 Pozzuoli, Italy; 3Department of Translational Medical Sciences, University of Naples Federico II, 80138 Napoli, Italy; 4Italian Society of Gerontology and Geriatrics (SIGG), 50129 Florence, Italy; 5Department of Life, Health and Environmental Sciences, University of L’Aquila, 67100 L’Aquila, Italy; 6San Raffaele Sulmona Institute, 67039 Sulmona, Italy

**Keywords:** BMI, crenotherapy, chronic low back pain

## Abstract

Crenotherapy is recognized as being effective in patients with osteoarthritis of the spine, but to date there is no indication if it is effective for patients who are overweight or obese. The aim of this study is to evaluate the efficacy of sulphurous crenotherapy on pain and disability in overweight/obese subjects affected by chronic low back pain from spine osteoarthritis. Forty-three patients (63 ± 8.8 years) affected by chronic low back pain from lumbar spine osteoarthritis were enrolled in this study. Subjects were treated with 2 weeks of sulphurous creno-treatments. Subjective pain was measured by a numerical rating scale score (NRS), and functional mobility of the lumbar spine was measured using the Oswestry Disability Index (ODI) before and after crenotherapy. Both crenotherapy groups (normal weight: A1; overweight/obese: A2) experienced significantly improved NRS and ODI scores (A1: *p* < 0.001 and *p* = 0.001; A2: *p* = 0.001 and *p* = 0.001). At end of the treatment, significant improvements were observed as a result of the crenotherapy in overweight/obese subjects in terms of pain reduction measured with NRS (*p* = 0.03) and in terms of function mobility of the lumbar spine measured with ODI (*p* = 0.006). This study highlights the beneficial effect of sulphurous crenotherapy on the painful symptomatology and disability in both normal weight and overweight/obese patients suffering from chronic low back pain associated with lumbar spine osteoarthritis.

## 1. Introduction

Lower back pain (LBP) affects 65–80% of the general population worldwide [1]. LBP very often evolves into chronic low back pain (CLBP) and represents a relevant social and economic burden as it is the fifth leading cause of medical care requests [1,2].

Several risk factors are described in the literature as causing lower back pain, including a previous episode of back pain, incorrect habits, such as maintaining a certain posture for long periods during day, overweight and obesity [3,4,5,6]. An association between body mass index (BMI) and LBP has been found in the U.S. population where LBP represents, together with osteoarthritis and fibromyalgia, the most frequent cause of chronic pain [7]. A recent study found a significant causal effect of BMI on both back pain and chronic back pain, with an increase in BMI conducing a 1.15 time greater chance of back pain [8].

Chronic low back pain (CLBP) is characterized by the presence of a persistent (more than 12 weeks) discomforting pain localized at the lumbar spine with or without irradiation to the buttocks and lower limbs associated with a functional limitation, considerable psychological stress, and insomnia [9,10].

Therefore, CLBP and obesity both affect quality of life (QoL), often-causing disability with deleterious social and economic consequences [11,12,13,14,15].

Analyzing pharmacotherapy, White et al. analyzed the safety and effectiveness of opioids, nonsteroidal anti-inflammatory drugs (NSAIDs), and antidepressants for the treatment of CLBP. These authors highlighted that NSAIDs are effective in CLBP but induce side effects with possibly dangerous consequences. They concluded that opioids and antidepressants may be considered but should not be routinely used for the treatment of CLBP [16]. A study investigating the beneficial effects of exercise training in patients suffering from hip osteoarthritis has shown that pain symptomatology was often present in individuals with higher BMI, and that obese patients had the highest pain levels, worse physical function, and strongly decreased QoL [17]. Recently, regarding CLBP in parallel with an updated Cochrane review on the efficacy of exercise, individual participant data (IPD) from 27 high-quality randomized clinical trials (including 3,514 participants) were recorded. Lower body mass index was also associated with better outcomes (reduced average pain and functional limitation) in the exercise groups compared with the no treatment/usual care groups [18].

Given the burden of the CLBP, particularly in subjects with higher BMI, researchers’ interest in possible therapeutic interventions for the treatment of this disease is now increasing. Among the available non-pharmacological therapeutic approaches, crenotherapy is considered a valid complementary therapy in patients with CLBP, as highlighted by recently updated reviews and meta-analyses [19,20,21] and by several clinical studies [22,23,24,25,26,27,28].

Our previous data [22] also showed that sulphurous crenotherapy improves pain perception, disability function, depression, and insomnia, which altogether affect the QoL of patients suffering from CLBP associated with lumbar spine osteoarthritis. In obese patients with knee osteoarthritis, hydrokinesitherapy in a thermal environment may result in pain relief, joint function improvement, and walking speed increase for 6 months until follow-up [29].

However, the efficacy of sulphurous crenotherapy in overweight/obese subjects affected by CLBP from spine osteoarthritis has not been investigated yet. Thus, there is an urgent need to test the feasibility of interventions that could be alternative or at least complementary to pharmacological therapy in overweight/obese subjects. The present study aims to investigate the efficacy of crenotherapy with sulphurous mineral water on pain and disability in overweight and obese patients suffering from CLBP with lumbar spine osteoarthritis, and to analyze whether the effects of sulphurous crenotherapy can vary between subjects with BMI within or above the normal range.

## 2. Materials and Methods

### 2.1. Study Population and Study Design

An observational study was conducted on 43 patients suffering from CLBP with lumbar spine osteoarthritis, who were referred to crenotherapy by their general practitioners. Inclusion criteria was the diagnosis of lumbar spine osteoarthritis (assessed by clinical and radiological examination) in the presence of CLBP with an NRS (Numerical Rating Scale) score > 3, lasting for at least 12 weeks.

The enrolled subjects were divided into four groups: Group A1 (normal BMI) and Group A2 (overweight/obese) received crenotherapy, Group B1 (normal BMI) and Group B2 (overweight/obese) received medical supervision with no crenotherapy. See Figure 1 for the research flow diagram.

The exclusion criteria were acute low back pain and/or radicular pain, intra-articular injections of steroids in the previous 3 months, recent surgical interventions, serious venous insufficiency, cancers, uncontrolled hypertension, diabetes, pregnancy, coronary heart disease, metal implants, cardiac pacemakers, and cognitive impairment.

### 2.2. Sulphurous Crenotherapy

Groups A1 and A2 (A1: *n* = 18, BMI = 23.7 ± 0.97; and A2: *n* = 25, BMI = 30.6 ± 4.2) underwent sulphurous crenotherapy, consisting of twelve applications of balneotherapy with medical sulphurous mineral water (which includes an appreciable amount of bicarbonate ions, 1.015 mg/L; calcium ions, 499 mg/L; and magnesium ions, 83.8 mg/L; in addition to bivalent sulphur and its compounds, 13.1 mg/L hydrogen sulfide; 6.2 pH) and peloids at the Terme di Telese Spa (Benevento, Italy).

This sulphurous crenotherapy cycle includes mud packs once a day, in the morning and preferably while the subject is fasting, applied to the lumbosacral region of the spine for 15 min at 44 °C, followed by a cleaning shower, and next by a bath in sulphurous mineral water at 38 °C for 10 min. Then, the patients, comfortably covered, rested for 15–20 min lying down or reclining.

This treatment also involved twelve applications of hydropinotherapy, which consisted of drinking one glass (of 250 mL) of sulphurous medical mineral water at room temperature once per day. The patients resided at home during the sulphurous crenotherapy. No patients used anti-inflammatory or muscle relaxant drugs, or performed physical therapy or exercises during the two weeks of sulphurous crenotherapy.

### 2.3. Control Groups

Twenty healthy age-matched volunteers were recruited from the waiting list as controls (Group B). Group B1 consisted of 10 subjects with BMI in the normal range, and group B2 of 10 subjects with BMI above the normal range. All controls did not perform physical therapy or exercises but had occasionally used NSAIDs during the two weeks of intervention under medical supervision. Their demographic characteristics are illustrated in Table 1.

### 2.4. Ethical Issues

The study obtained the approval of the Ethics Committee (No. 7 r.p.s.o./2020) Campania Sud, Naples, Italy, according to the Declaration of Helsinki and its amendments. All participants provided their informed consent.

### 2.5. Measurements

Low back pain and disability function were evaluated before and after sulphurous crenotherapy, stratifying for BMI (Group A1: *n* = 18, BMI = 23.7 ± 0.97; Group A2: *n* = 25, BMI = 30.6 ± 4.2). BMI, which combines a person’s weight with their height, is a screening tool that can indicate whether a person is underweight, has a healthy weight, overweight, or obese.

Lower back pain was measured using the numerical rating scale (NRS) score, which ranges from 0 to 10 (score 0 = absent pain, score 1–3 = mild pain, score 4–6 = moderate pain, score 7–10 = severe pain) [30,31].

The analysis of disability and function at the lumbar spine was performed using the Italian version of the Oswestry Disability Index (ODI) [32,33], a tool commonly used to quantify a permanent functional disability. Total scores range from 0 to 50 and are converted to percentages from 0 to 100%: 0 corresponds to no disability and 100 to the maximum disability. According to Fairbank and Pynent [33], we considered five levels of scoring the disability: minimal (0–20%), moderate (21–40%), severe (41–60%), crippled (61–80%) and exaggerated or bedbound disability (81–100%). The occurrence of undesired events and/or adverse reactions during the crenotherapy was recorded in a case report form (CRF).

### 2.6. Statistical Analysis

The results are expressed as mean ± standard deviation (SD). The inferential statistics was performed by applying non parametric tests. A Kruskal–Wallis analysis was performed to compare the data of the four groups of subjects, followed by post hoc analysis performed by means of the Mann–Whitney u-test. The within-group differences between the first and the second evaluation were assessed using the Wilcoxon test. The alpha-level of significance was set at 0.05 for primary analyses and at 0.025 for post hoc analysis according to Bonferroni correction, which was performed to avoid the possible inflation of first type errors. The statistical analysis was performed using IBM SPSS Statistics, Version 23.0. NY, USA. 

## 3. Results

The study population consisted of 43 patients (A group) subdivided into two subgroups (A1 and A2) on the basis of their BMI. The main demographic characteristics of the sample are summarized in the Table 1. Before sulphurous crenotherapy, the patients of the A2 subgroup presented a more severe lumbar pain symptomatology than the A1 subgroup, as shown by the NRS score (A2 subgroup: 5.36 ± 3.0; A1 subgroup: 4.2 ± 2.7) (Table 2), with a greater negative impact on lumbar functional mobility as demonstrated by the ODI (A1 subgroup: 13% ± 0.08; A2 subgroup: 19% ± 0.13) (Table 3). At the end of the sulphurous crenotherapy, there was a significant (*p* < 0.05) reduction in both painful symptomatology and functional disability of the lumbar spine in both subgroups vs. baseline, as demonstrated by the values of the NRS score (A1 subgroup: 1.7 ± 1.9; A2 subgroup: 2.92 ± 2.4) (Table 2) and ODI (A1 subgroup: 8% ± 0.08; A2 subgroup: 12% ± 0.10) (Table 3). No significant changes were observed in control subgroups (B1 and B2). In Figure 2, the prevalence of the ODI categories before and after crenotherapic treatment is shown.

The Kruskal–Wallis analysis revealed the absence of significant differences among the four subgroups (A1, A2, B1, B2) at the first assessment in terms of NRS scores (*p* = 0.597), yet conversely highlighted a statistically significant difference at the second assessment (*p* = 0.003). Post hoc analysis showed that this was due to the difference between A1 and B1 (*p* = 0.005), more so than the difference between A2 and B2 (*p* = 0.031, not significant after Bonferroni correction). The Wilcoxon test showed significant improvements from first to second assessment for A1 (*p* < 0.001 and *p* = 0.001 for NRS and ODI scores, respectively) and A2 (*p* = 0.001 and *p* = 0.001), but neither for B1 (*p* = 0.285 and *p* = 0.655) nor for B2 (*p* = 0.039 and *p* = 0.246, not significant after Bonferroni correction) (Figure 3 and Figure 4).

In terms of ODI scores, the Kruskal–Wallis analysis showed significant differences at the first assessment (*p* = 0.032) and increased differences at the second (*p* = 0.004). Post hoc analysis showed that the difference pre-intervention was due to the difference between B1 and B2 (*p* = 0.023), whereas post-intervention was due to the difference between A2 and B2 (*p* = 0.006) and not to that of A1 and B1 (*p* = 0.226) (Figure 4). All the subjects in the A groups completed the 2-week crenotherapy. In the subjects enrolled in all groups, no undesired events were reported.

## 4. Discussion

This study aimed to investigate the efficacy of sulphurous crenotherapy on pain and functional mobility of overweight/obese subjects affected by CLBP, and evaluate the subjects’ response to sulphurous crenotherapy in comparison to normal weight subjects affected by CLBP.

Smuck et al. [10] have demonstrated that high BMI is a risk factor for LBP the in the U.S. population. LBP frequently evolves to in CLBP in the presence of risk factors, such as lumbar spine osteoarthritis with a BMI above the normal range. The latter might prejudice the musculoskeletal health with negative impact on the QoL as demonstrated in overweight/obese individuals [34,35,36,37]. CLBP leads to functional limitations and significant restrictions on daily activities [38]. With regard to health-related quality of life (HRQL), patients affected by CLBP experience a negative impact in different life domains, such as physical and mental wellbeing, social relationships, and functional ability [18]. It has been postulated that obesity causes back pain through a constant mechanical load on the spine and systemic chronic inflammation. Literature data shows that increased joint inflammation, osteoarthritis, and an excess of oxidized low-density lipoproteins leads to an increase in pro-inflammatory cytokines and their receptors, thereby creating a vicious circle in which there is a link between acute/chronic inflammation and a BMI above the normal range [39,40].

Our study highlights the beneficial effects (and the consequential improvement to QoL) of the crenotherapy cycle with medical sulphurous mineral water (consisting of balneotherapy cycle and peloids, in addition to sulphurous drinking therapy) on the persistent disabling painful symptomatology that characterizes patients with CLBP associated with lumbar spine osteoarthritis, which is in accordance with the literature [20,21,22,23,24,25].

Our results are supported by the analysis of the distribution of the ODI categories. Indeed, at the end of the sulphurous crenotherapy in both groups, there was a disappearance of disability or a change from categories of greater to lower disability (Figure 1). Notably, no patients with normal BMI showed a severe ODI category before the treatment. The therapeutic effects of such crenotherapic treatment are exemplified by analgesic (with an increase in β-endorphin serum levels) [41], muscle relaxant (with an increase in the extensibility of collagen tissue, thereby counteracting ankyloses and fibrotic retraction with decreased stiffness), trophic and vasodilator effects. The analgesic and muscle relaxant effects are able to break the vicious cycle: pain-muscular contracture-altered dynamic-pain, which characterized chronic arthropathies [42,43], allowing a better use of the joints. In osteoarthritis, an important role is played also by the release of free radical species produced by synovial inflammatory cells and by the chondrocytes [44,45]. In this context sulphurous crenotherapy seems to have an antioxidant role, as suggested by several studies. Several pieces of evidence have shown a protective effect favored by reducing the properties of the sulfhydryl group contained in the sulphurous mineral water against oxidative DNA damage associated with inflammatory respiratory diseases [46]. Moreover, antioxidant and hypoglycemic effects of hydrogen sulphide (H_2_S) in patients affected by metabolic and/or gastrointestinal disorders have been demonstrated [47].

Our study sheds light on the fact that patients suffering from CLBP with lumbar spine osteoarthritis who are overweight/obese have a NRS pain score higher than those who are normal weight, which is in line with important aggregate individual patient data [18]. A significant association between high BMI values and LBP has been observed in individuals from several countries, such as Finland, Poland, Spain, Russia, and South Africa [34].

Interestingly, our research highlighted that sulphurous crenotherapy is able to ameliorate lumbar pain and functional mobility in such overweight/obese subjects. However, the reduction was more consistent in patients with normal BMI, with a 60% reduction of pain symptomatology, when compared with overweight/obese patients who demonstrated a 46% reduction, despite all subjects undergoing the same sulphurous crenotherapic cycle of equal intensity and duration. This is an important result because it validates the claim that crenotherapy can be considered a valid therapeutic option for back pain and disabilities resulting from osteoarthritis, probably thanks to its antioxidant effect that can modulate patients’ anti-inflammatory status, which is particularly high in overweight and obese subjects.

In line with our results, Karagulle et al. in their review have highlighted that balneotherapy was superior to long-term tap water therapy in relieving pain and improving function, and that spa therapy combining balneotherapy with mud pack therapy and/or exercise therapy, physiotherapy, and/or education, was effective in the management of lower back pain, and is also superior or equally effective to both short- and long-term control treatments [48].

Nguyen et al. [49] in their study showed that patients with osteoarthritis of the hip/knee or lumbar spine, who underwent a crenotherapic cycle of balneotherapy, showed improvement of pain, functional impairment, and QoL with a reduced intake of symptomatic drugs (NSAID and analgesic drugs) after the 6 month follow-up period.

This study has some limitations which deserve consideration. The first limitation of this study is the sue pf patients on a waiting list as the control group, as this might underestimate the effectiveness of the control intervention. A second limitation is the possible beneficial placebo effect related to the spa environment. However, these patients stayed in the spa only for the time period of the crenotherapy cycle. A third limitation is the small sample size of the study population. Further larger studies with a randomized controlled design are needed to confirm our results.

## 5. Conclusions

In conclusion, our research suggests that crenotherapy with the use of sulphur medicinal waters in the form of peloid, bath, and drink, may represent a useful treatment for pain reduction and function improvement in overweight/obese patients suffering from CLBP associated to lumbar spine osteoarthritis. In addition, our study confirms the positive effect of crenotherapy in normal weight subjects affected by chronic low back pain caused by lumbar spine osteoarthritis.

## Figures and Tables

**Figure 1 healthcare-10-01800-f001:**
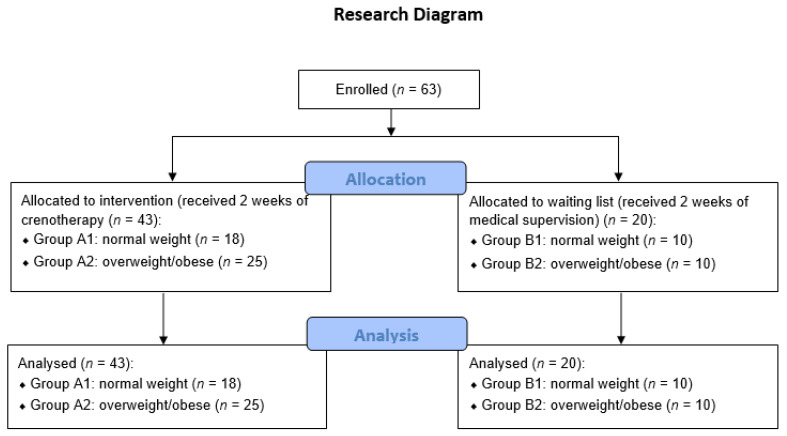
Research flow diagram.

**Figure 2 healthcare-10-01800-f002:**
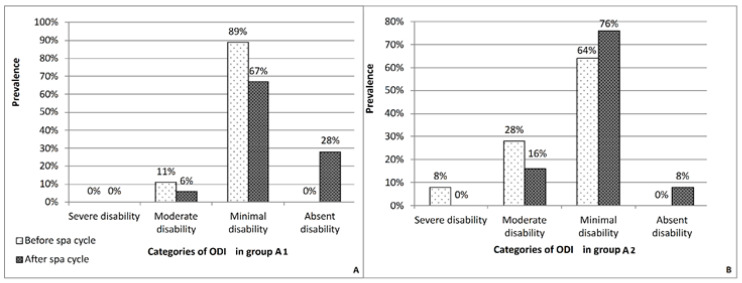
Prevalence of Oswestry Disability Index (ODI) categories before and after sulphurous crenotherapy. (**A**): A1 subgroup; (**B**): A2 subgroup.

**Figure 3 healthcare-10-01800-f003:**
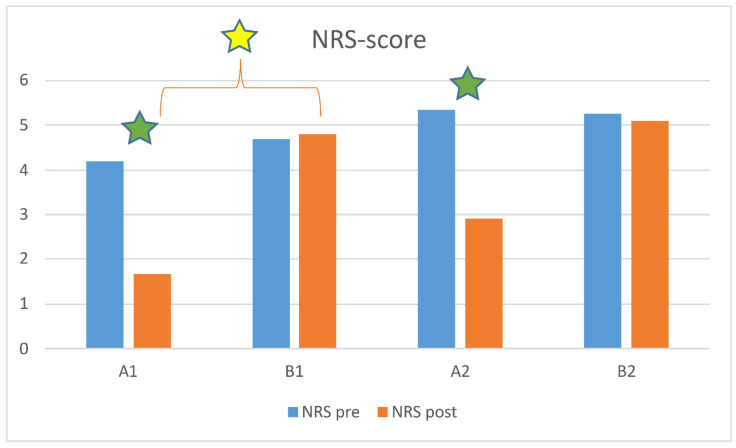
NRS score between the four subgroups: A1 (spa treatment, normal BMI), A2 (spa treatment, overweight/high BMI), B1 (no spa treatment, normal BMI), and B2 (no spa treatment, overweight/high BMI). Green stars indicate a statistically significant difference between in pre- and post-analysis results (*p* < 0.005); yellow stars indicate a statistically significant difference between groups (*p* < 0.0025; Bonferroni correction).

**Figure 4 healthcare-10-01800-f004:**
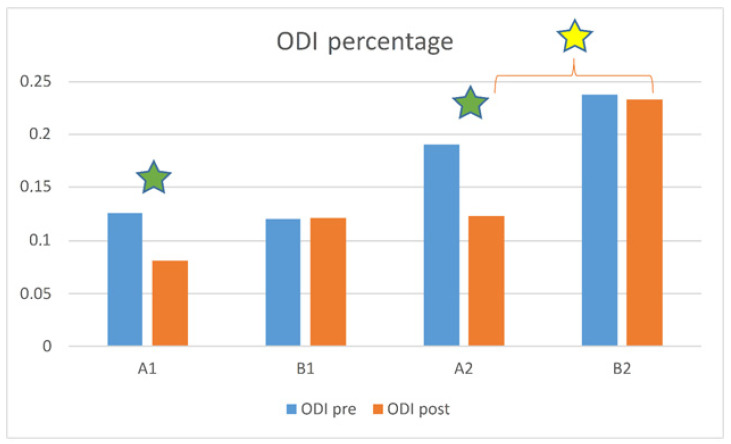
ODI score between the four subgroups: A1 (spa treatment, normal BMI), A2 (spa treatment, overweight/high BMI), B1 (no spa treatment, normal BMI), and B2 (no spa treatment, overweight/high BMI). Green stars indicate a statistically significant difference between in pre- and post-analysis results (*p* < 0.005); yellow stars indicate a statistically significant difference between groups (*p* < 0.0025; Bonferroni correction).

**Table 1 healthcare-10-01800-t001:** Demographic characteristics of the A (*n* = 43, treated with sulphurous crenotherapy) and B (*n* = 20, control) groups; and the A1 and B1 subgroups, with normal weight (18.5 ≤ BMI ≤ 24.99 kg/m^2^), and A2 and B2 subgroups, overweight/obese (BMI ≥ 25 kg/m^2^), at baseline.

	Age (Years) Mean ± SD	Range Age (Years)	Median	Female %	Male %	BMI * Mean ± SD
Study group (*n* = 43)	63 ± 8.8	41–81	62	58	42	27.7 ± 4.7
A1 subgroup (*n* = 18)	63 ± 8.9	44–81	62	37	33	23.7 ± 0.97
A2 subgroup (*n* = 25)	62 ± 8.9	41–77	61	52	48	30.6 ± 4.2
Control group (*n* = 20)	63 ± 11	46–83	62	60	40	26.6 ± 5.3
B1 subgroup (*n* = 10)	61 ± 8.8	47–80	59	70	30	22.3 ± 1.9
B2 subgroup (*n* = 10)	65.5 ± 13.2	46–83	65.5	50	50	30.8 ± 3.9

* BMI = weight (kg)/ height (m^2^).

**Table 2 healthcare-10-01800-t002:** Total NRS (numeric rating scale) pain score (mean ± SD) score for the four subgroups: A1 (spa treatment, normal BMI), A2 (spa treatment, overweight/high BMI), B1 (no spa treatment, normal BMI), and B2 (no spa treatment, overweight/high BMI).

	Total NRS Score		
Subgroup (*n*)	Before	After	*p*-Value
A1 (18)	4.2 ± 2.7	1.7 ± 1.9	<0.001
A2 (25)	5.36 ± 3.0	2.92 ± 2.4	0.0005
B1 (10)	4.7 ± 3.1	4.8 ± 3.0	0.4
B2 (10)	5.3 ± 2.7	5.1 ± 2.7	0.1

**Table 3 healthcare-10-01800-t003:** Total ODI (Oswestry Disability Index) score (mean ± SD) for the four subgroups: A1 (spa treatment, normal BMI), A2 (spa treatment, overweight/high BMI), B1 (no spa treatment, normal BMI), and B2 (no spa treatment, overweight/high BMI).

Subgroup (*n*)	Total ODI Score	Mean ± SD	*p*-Value (Wilcoxon Test)
	Before	After	
A1 (18)	13% ± 0.08	8% ± 0.08	*p* < 0.001
B1 (10)	12.05% ± 0.1	12.10% ± 0.1	*p* > 0.05
A2 (25)	19% ± 0.13	12% ± 0.10	*p* = 0.001
B2 (10)	23.8% ± 0.1	23.3% ± 0.1	*p* > 0.05

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
