# Peer review of "Sulphurous Crenotherapy Is Effective at Reducing Pain and Disability in Overweight/Obese Patients Affected by Chronic Low Back Pain from Spine Osteoarthritis"

_healthcare, 2022, doi:10.3390/healthcare10091800_

Round 1
Reviewer 1 Report
The authors should give information about the limitations of the study.
The authors should also give details about the therapies such as physical therapy or exercise if they applied.
The discussion can be improved adding some therapies have been used for chronic low back pain in the related literature and comparing their results.
In conclusion section, the authors should their recommendations for the clinicians working in this field more clearly.
Author Response
We thank the Editor and the reviewers for the opportunity to revise our manuscript.
Please find a version of the manuscript, changed following all requests and suggestions.
All changes are in red throughout the entire manuscript.
Reviewer1#
The authors should give information about the limitations of the study.
AU Replay: We would thank the reviewer for the helpful comments.
Following her/his suggestions we implemented the description of limitations as follows (see Discussion section).
“This study presents some limitations which deserve consideration. The first limitation is the waiting list as control group which might underestimate the effectiveness of the control intervention; second limitation is the possible beneficial placebo effect in the patient’s affected sulphurous crenotherapy related to the attendance of the Spa, although these patients attended the Spa only for crenotherapies; third limitation is the small sample size of each group, whereas in a similar study, the recruited population has a similar or smaller size. ”
The authors should also give details about the therapies such as physical therapy or exercise if they applied.
AU Reply: We would thank the reviewer for the suggestions. In the methods we highlighted the possible applied of the use of other therapies. (see 2.2 Sulphurous Crenotherapy section and 2.3. Control groups section)
“2.2. Sulphurous crenotherapy
The group A (A1: n=18, BMI=23.7±0.97 and A2: n=25, BMI=30.6±4.2) underwent to sulphurous crenotherapy, consisting in 12 applications of balneotherapy with medical sulphurous mineral water (which includes an appreciable amount of bicarbonate ions, 1.015mg/L; calcium ions, 499mg/L; and magnesium ions, 83.8mg/L; in addition to bivalent Sulphur and its compounds, 13.1 mg/L hydrogen sulfide; 6.2 pH) and peloids at the Terme di Telese S.p.A. (Benevento-Italy).
This sulphurous crenotherapy cycle includes of mud packs one a day, in the morning and preferably in fasting condition, applied to the lumbosacral region of the spine for 15 min. at 44°C, followed by a cleaning shower, and a bath in sulphurous mineral water at 38°C for 10 min. Then, the patients, comfortably covered, rested for 15-20 min lying down or reclined.
This treatment was associated to 12 applications of hydropinotherapy, which consisted in drinking daily 1 glass (of 250mL) of sulphurous medical mineral water of Telese Terme S.p.A., at room temperature. The patients resided at home during the sulphurous crenotherapy. No patients used anti-inflammatory or muscle relaxant drugs, or performed physical therapy or exercises during the 2-weeks sulphurous crenotherapy.
2.3. Control groups
Twenty healthy age-matched volunteers were recruited from the waiting list as controls (Group B). All participants were Caucasians. Group B1 consisted of 10 subjects with BMI in the normal range, and group B2 of 10 subjects with BMI above the normal range. All controls did not perform physical therapy or exercises, but had occasionally used NSAIDs in the 2-weeks of intervention under medical supervision. Their demographic characteristics are illustrated in Table 1.”
The discussion can be improved adding some therapies have been used for chronic low back pain in the related literature and comparing their results.
AU Reply: We would thank the reviewer for the suggestions. In the Discussion section we have inserted the Suggestions as follow.
“This is an important result because crenotherapy should be considered a valid therapeutic option for back pain and disabilities from osteoarthritis, probably thanks to its antioxidant effect that might modulate patients’ anti-inflammatory status that is particular high in overweight and obese subjects.
In line with our results, Karagulle et al in a review have highlighted that balneotherapy was superior in long term to tap water therapy in relieving pain and improving function and that spa therapy combining balneotherapy with mud pack therapy and/or exercise therapy, physiotherapy, and/or education was effective in the management of low back pain and superior or equally effective to the control treatments in short and long terms [50].
Nguyen et al [51] in patients with osteoarthritis of the hip/knee or lumbar spine, that effected crenotherapic cycle of the balneotherapy, showed improvement of pain, functional impairment and QoL with a reduced intake of symptomatic drugs (NSAID and analgesic drugs) after 6 month follow-up period.
This study presents some limitations which deserve consideration. The first limitation is the waiting list as control group which might underestimate the effectiveness of the control intervention; second limitation is the possible beneficial placebo effect in the patient’s affected sulphurous crenotherapy related to the attendance of the Spa, although these patients attended the Spa only for crenotherapies; third limitation is the small sample size of each group, whereas in a similar study, the recruited population has a similar or smaller size.”
In conclusion section, the authors should their recommendations for the clinicians working in this field more clearly.
Reply: Thanks to the reviewer for the comment. Following her/his suggestion we have made the changes, (see Conclusion section)
“In conclusion, our research suggests that the sulphurous crenotherapy, including the associated use of sulphur medicinal waters in the form of peloid, bath and drink, may represent a useful treatment for pain reduction and function improvement also in overweight/obese patients suffering from CLBP associated to lumbar spine osteoarthritis. In addition, our study confirms the positive effect of crenotherapy in normal weight subjects affected by chronic low back pain caused by lumbar spine osteoarthritis.
No doubt, further studies with a larger sample size with randomized controlled design are needed to confirm our results and suggestions.”
We sincerely thank the Reviewer.
- Conclusions
In conclusion, our research suggests that the sulphurous crenotherapy, including the associated use of sulphur medicinal waters in the form of peloid, bath and drink, may represent a useful treatment for pain reduction and function improvement also in overweight/obese patients suffering from CLBP associated to lumbar spine osteoarthritis. In addition, our study confirms the positive effect of crenotherapy in normal weight subjects affected by chronic low back pain caused by lumbar spine osteoarthritis.
No doubt, further studies with a larger sample size with randomized controlled design are needed to confirm our results and suggestions.
Reviewer 2 Report
This study aims to evaluate the efficacy of sulphur spa therapy on pain and disability in overweight/obese subjects affected by chronic low back pain due to osteoarthritis of the spine. Studies are needed to evaluate the role of balneotherapy and thermal medicine in musculoskeletal pathology and especially in low back pain. The authors found a beneficial effect of muddy sulphurous water intervention on pain symptomatology and disability in both normal weight and overweight/obese patients suffering from chronic low back pain associated with osteoarthritis of the lumbar spine.
To me, this paper is suited for publication, after some clarification and modification from the authors.
- I recommend that you review the expression "spa" in the sense of "salus per aqua" as it is not correct. I advise you to use the terminology recommended in Guterbrunner's article whose citation is attached.
Gutenbrunner C et al. Int J Biometeorol. 2010 Sep;54(5):495-507
- In the methods section you should clarify and/or expand on the following aspects:
o The authors explain that the study was conducted in 43 Caucasian patients suffering from CLBP with osteoarthritis of the lumbar spine, who were referred by their general practitioners (GPs) (group A) to mud bath therapy at the spa, but do not indicate anything about the selection. of the 20 patients in the control group, and why there has been no randomisation.
o Did the patients reside at the spa or at home during the treatment. You should specify.
o Was the evaluation of groups A and B performed by the same investigator?
- The occurrence of unwanted events and/or adverse reactions during the spa treatment are not reflected in the results.
Author Response
We thank the Editor and the reviewers for the opportunity to revise our manuscript.
Please find a version of the manuscript, changed following all requests and suggestions.
All changes are in red throughout the entire manuscript.
Reviewer2#
This study aims to evaluate the efficacy of sulphur spa therapy on pain and disability in overweight/obese subjects affected by chronic low back pain due to osteoarthritis of the spine. Studies are needed to evaluate the role of balneotherapy and thermal medicine in musculoskeletal pathology and especially in low back pain. The authors found a beneficial effect of muddy sulphurous water intervention on pain symptomatology and disability in both normal weight and overweight/obese patients suffering from chronic low back pain associated with osteoarthritis of the lumbar spine.
To me, this paper is suited for publication, after some clarification and modification from the authors.
- I recommend that you review the expression "spa" in the sense of "salus per aqua" as it is not correct. I advise you to use the terminology recommended in Guterbrunner's article whose citation is attached.
Gutenbrunner C et al. Int J Biometeorol. 2010 Sep;54(5):495-507
AU Reply: Thanks to the reviewer for the comment.
Following her/his suggestion we have made the changes, highlighted in red, regarding the expression "spa" (see Article) with insertion of the reference Gutenbrunner C, Bender T, Cantista P, Karagülle Z. A proposal for a worldwide definition of health resort medicine, balneology, medical hydrology and climatology. Int J Biometeorol. 2010 Sep;54(5):495-507. doi: 10.1007/s00484-010-0321-5. Epub 2010 Jun 9. PMID: 20532921. (see References section, n. 19)
- In the methods section you should clarify and/or expand on the following aspects:
o The authors explain that the study was conducted in 43 Caucasian patients suffering from CLBP with osteoarthritis of the lumbar spine, who were referred by their general practitioners (GPs) (group A) to mud bath therapy at the spa, but do not indicate anything about the selection. of the 20 patients in the control group, and why there has been no randomisation.
Reply: We would thank the reviewer for the suggestions. We have now addedd this important information in method section as follow:
“2.2. Sulphurous crenotherapy
The group A (A1: n=18, BMI=23.7±0.97 and A2: n=25, BMI=30.6±4.2) underwent to sulphurous crenotherapy, consisting in 12 applications of balneotherapy with medical sulphurous mineral water (which includes an appreciable amount of bicarbonate ions, 1.015mg/L; calcium ions, 499mg/L; and magnesium ions, 83.8mg/L; in addition to bivalent Sulphur and its compounds, 13.1 mg/L hydrogen sulfide; 6.2 pH) and peloids at the Terme di Telese S.p.A. (Benevento-Italy).
This sulphurous crenotherapy cycle includes of mud packs one a day, in the morning and preferably in fasting condition, applied to the lumbosacral region of the spine for 15 min. at 44°C, followed by a cleaning shower, and a bath in sulphurous mineral water at 38°C for 10 min. Then, the patients, comfortably covered, rested for 15-20 min lying down or reclined.
This treatment was associated to 12 applications of hydropinotherapy, which consisted in drinking daily 1 glass (of 250mL) of sulphurous medical mineral water of Telese Terme S.p.A., at room temperature. The patients resided at home during the sulphurous crenotherapy. No patients used anti-inflammatory or muscle relaxant drugs, or performed physical therapy or exercises during the 2-weeks sulphurous crenotherapy.
2.3. Control groups
Twenty healthy age-matched volunteers were recruited from the waiting list as controls (Group B). All participants were Caucasians. Group B1 consisted of 10 subjects with BMI in the normal range, and group B2 of 10 subjects with BMI above the normal range. All controls did not perform physical therapy or exercises, but had occasionally used NSAIDs in the 2-weeks of intervention under medical supervision. Their demographic characteristics are illustrated in Table 1. “
o Did the patients reside at the spa or at home during the treatment. You should specify.
AU Reply: Thanks to the reviewer for the suggestion. Patients reside at home, we have added this information in the manuscript (see Materials and Methods section).
o Was the evaluation of groups A and B performed by the same investigator?
AU Reply: Yes the evaluation of groups A and B has been performed by the same investigator, we have added this information in the manuscript (see Materials and Methods section).
- The occurrence of unwanted events and/or adverse reactions during the spa treatment are not reflected in the results.
AU Reply: We would thank the reviewer for the comment. We implemented this description (see Results section).
We sincerely thank the Reviewer.
Reviewer 3 Report
This study has its own scarcity as it saw the effect of sulfurous spa therapy in patients with spine OA. However, the contents below need to be revised a lot.
Abstract
In the Background, prior studies related to this study should be mentioned. The content described by the author corresponds to the purpose of this study.
Introduction
Line 50-51: Reference no. 9 seems to be more related to the sentence corresponding to line 52-53.
Line 52-53: It is necessary to correct the distinction of paragraphs.
Line 74-75: There are studies on mud-bath spa and CLBP (including the authors' papers). The differences between the preceding studies presented below and this study should be emphasized in the relevant paragraph.
Nguyen, M., Revel, M., & Dougados, M. (1997). Prolonged effects of 3 week therapy in a spa resort on lumbar spine, knee and hip osteoarthritis: follow-up after 6 months. A randomized controlled trial. British journal of rheumatology, 36(1), 77-81.
Costantino, M., Conti, V., Corbi, G., Marongiu, F., Marongiu, M. B., & Filippelli, A. (2019). Sulphurous mud-bath therapy for treatment of chronic low back pain caused by lumbar spine osteoarthritis. Internal and Emergency Medicine, 14(1), 187-190.
Methods
- Was the normality test conducted between groups in this study? The results shall be inserted in Table 1. In the paired t-test, a normality test should be performed based on the “post – pre” value.
- Figure 1, 2: The pixel of the picture is too low. It is necessary to change the file to a higher resolution.
- It is necessary to add a research diagram for readers' understanding.
Discussion
In the discussion section, the author's thoughts should be described with appropriate grounds for the reasons for the results of this study. However, it seems rather meager in this part. It is necessary to further explain the mechanism of antioxidant and anti-inflammatory, or to further persuade the results of this study with other evidence.
Conclusions
Line 282-283: This sentence should be described at the end of the Discussion part. In addition, the limitations of this study that the author thinks should be added to the discussion.
Author Response
We thank the Editor and the reviewers for the opportunity to revise our manuscript.
Please find a version of the manuscript, changed following all requests and suggestions.
All changes are in yellow throughout the entire manuscript.
Reviewer3#
Abstract
In the Background, prior studies related to this study should be mentioned. The content described by the author corresponds to the purpose of this study.
AU Replay: thanks for the indication. We have now add in the abstract the following sentence: “Crenotherapy is recognized as effective in patients with osteoarthritis of the spine but to date there is no indication in patients with overweight and obesity.”
Introduction
Line 50-51: Reference no. 9 seems to be more related to the sentence corresponding to line 52-53.
Line 52-53: It is necessary to correct the distinction of paragraphs.
AU Reply: Thanks to the reviewer for the comment. Following her/his suggestion we have made the changes (see References section, n. 9,10 and 11-15)
Line 74-75: There are studies on mud-bath spa and CLBP (including the authors' papers). The differences between the preceding studies presented below and this study should be emphasized in the relevant paragraph.
Costantino, M., Conti, V., Corbi, G., Marongiu, F., Marongiu, M. B., & Filippelli, A. (2019). Sulphurous mud-bath therapy for treatment of chronic low back pain caused by lumbar spine osteoarthritis. Internal and Emergency Medicine, 14(1), 187-190. (Reference n. 22)
Nguyen, M., Revel, M., & Dougados, M. (1997). Prolonged effects of 3 week therapy in a spa resort on lumbar spine, knee and hip osteoarthritis: follow-up after 6 months. A randomized controlled trial. British journal of rheumatology, 36(1), 77-81. (Reference n. 51)
AU Repley: Thanks to the reviewer for the comment. Following her/his suggestion we have made the
following integration (see Introduction section, and see discussion section)
“ Introduction
Our previous data [22] also showed that sulphurous crenotherapy improves pain perception, disability function, depression and insomnia that all together affect the QoL of patients suffering from CLBP associated with lumbar spine osteoarthritis. In obese pa-tients with knee osteoarthritis hydrokinesitherapy in thermal environment may determine pain relief, joint function improvement, and walking speed increase until 6 months of fol-low-up [29].
However the efficacy of sulphurous crenotherapy in overweight/obese subjects af-fected by CLBP from spine osteoarthritis has not been investigated yet...”
“Discussion
Nguyen et al [51] in patients with osteoarthritis of the hip/knee or lumbar spine, that effected crenotherapic cycle of the balneotherapy, showed improvement of pain, function-al impairment and QoL with a reduced intake of symptomatic drugs (NSAID and analge-sic drugs) after 6 month follow-up period.”
Methods
- Was the normality test conducted between groups in this study? The results shall be inserted in Table 1. In the paired t-test, a normality test should be performed based on the “post – pre” value.
AU Reply: Thanks for highlight this point: we have not computed the normality test, because, given the small number of subjects for some subgroups, it could be difficult to reject the null hypothesis of normality. Furthermore we also had ordinal scores of clinical scales for which non parametric analyses are more suitable than parametric ones that should be used with continuous measures. Hence we preferred to use non-parametric tests that provide more conservative (and hence reliable) results than their parametric counterparts.
- Figure 1, 2: The pixel of the picture is too low. It is necessary to change the file to a higher resolution.
- It is necessary to add a research diagram for readers' understanding.
AU Reply: Thanks for the indication, we have addedd a reserch diagram, and we have improved the quality of the images.
Discussion
In the discussion section, the author's thoughts should be described with appropriate grounds for the reasons for the results of this study. However, it seems rather meager in this part. It is necessary to further explain the mechanism of antioxidant and anti-inflammatory, or to further persuade the results of this study with other evidence.
Reply: We would thank the reviewer for the suggestions. In the Discussion section, we have inserted the suggestions (see Discussion section) modifying the test as follow:
“]. In this context sulphurous crenotherapy seems to have an antioxidant role as suggested by several studies, that have shown a protective effect favored by reducing properties of the sulfhydryl group contained in the sulphurous mineral water against oxidative DNA damage associated with inflammatory respiratory diseases [47], an antioxidant and hy-poglicemic effects of hydrogene sulphide (H2S) in patients affected by metabolic and/or gastrointestinal disorders [48].
Vaamonde-García et al. showed in an experimental model of osteoarthritis reduced cartilage destruction and oxidative damage as a result of sulphurous balneotherapy [49].
In the present study, we have shed light that among patients suffering from CLBP with lumbar spine osteoarthritis, those who were overweight/obese subjects showed a NRS pain score higher than those who were normal weight in line with most important aggregate individual patient data [18]. A significant association between high BMI values and LBP has been observed in individuals from several countries, such as Finland, Po-land, Spain, Russia, and South Africa [34].
Interestingly, our research highlighted that sulphurous crenotherapy considered is able to ameliorate lumbar pain and functional mobility in such overweight/obese subjects on respect control subjects. However, the reduction was more consistent in patients with normal BMI, with a 60% reduction of pain symptomatology, when compared with over-weight/obese patients who demonstrated a 46% reduction, despite all subjected under-went the same sulphurous crenotherapic cycle of equal intensity and duration. This is an important result because crenotherapy should be considered a valid therapeutic option for back pain and disabilities from osteoarthritis, probably thanks to its antioxidant effect that might modulate patients’ anti-inflammatory status that is particular high in overweight and obese subjects.”
Conclusions
Line 282-283: This sentence should be described at the end of the Discussion part. In addition, the limitations of this study that the author thinks should be added to the discussion.
AU Reply: We would thank the reviewer for the helpful comments. Following her/his suggestions we implemented the description of limitations as follow
“……This study presents some limitations which deserve consideration. The first limitation is the waiting list as control group which might underestimate the effectiveness of the control intervention; second limitation is the possible beneficial placebo effect in the patient’s affected sulphurous crenotherapy related to the attendance of the Spa, although these patients attended the Spa only for crenotherapies; third limitation is the small sample size of each group, whereas in a similar study, the recruited population has a similar or smaller size.
- Conclusions
In conclusion, our research suggests that the sulphurous crenotherapy, including the associated use of sulphur medicinal waters in the form of peloid, bath and drink, may represent a useful treatment for pain reduction and function improvement also in overweight/obese patients suffering from CLBP associated to lumbar spine osteoarthritis. In addition, our study confirms the positive effect of crenotherapy in normal weight subjects affected by chronic low back pain caused by lumbar spine osteoarthritis.
No doubt, further studies with a larger sample size with randomized controlled design are needed to confirm our results and suggestions.
We sincerely thank the Reviewer.
Round 2
Reviewer 3 Report
- Line 50-53 and Line 54-55 shall be written in one paragraph.
- Line 75-80, Line 81-82, and Line 83-85 shall be written in one paragraph.
- Line 81: However -> However,
- In Figure 1, since there are no dropouts from this study, it seems that it is not necessary to include “End of treatment”.
- The resolution of Figure 2-4 has not been improved.
- Line 262-267: This sentence is too long and grammatically incorrect. It must be corrected.
- Line 269-270: This sentence should be included within the appropriate paragraph. A sentence should not be a paragraph.
- Line 287-292 and Line 293-296 shall be written in one paragraph.
- Line 297-303: The first limitation is ~. Second, this study is the possible ~. Third, ~~.
I think it would be better to fill it out as above.
- Line 311-312: This sentence should be deleted from the conclusion part and moved behind line 303.
- It is necessary to confirm that the revised part of this study is grammatically correct.
Author Response
We thank the Editor and the reviewers for the opportunity to revise our manuscript.
Please find a version of the manuscript, changed following all requests and suggestions.
- Line 50-53 and Line 54-55 shall be written in one paragraph.
Reply: ok thank
- Line 75-80, Line 81-82, and Line 83-85 shall be written in one paragraph.
Reply: ok thanks. We made the change.
- Line 81: However -> However,
Reply: ok thanks. We made the change.
- In Figure 1, since there are no dropouts from this study, it seems that it is not necessary to include “End of treatment”.
Reply: Thanks to the reviewer for the suggestion. We made it.
- The resolution of Figure 2-4 has not been improved.
Reply: Thanks to the reviewer for the suggestion. We improved the resolution of figure 2-4.
- Line 262-267: This sentence is too long and grammatically incorrect. It must be corrected
Reply: ok thanks. We rephrased the sentences.
- Line 269-270: This sentence should be included within the appropriate paragraph. A sentence should not be a paragraph.
Reply: We beg pardon. We deleted the sentence and corrected the references properly.
- Line 287-292 and Line 293-296 shall be written in one paragraph.
Reply: ok thanks. We made the change.
- Line 297-303: The first limitation is ~. Second, this study is the possible ~. Third, ~~.
I think it would be better to fill it out as above.
Reply: Thanks to the reviewer for the comment. Following her/his suggestion we have made the changes (see Discussion section)
- Line 311-312: This sentence should be deleted from the conclusion part and moved behind line 303.
Reply: Thanks to the reviewer for the comment
- It is necessary to confirm that the revised part of this study is grammatically correct.
Reply: The manuscript is revised by two experts and is grammatically correct (All later changes are in yellow throughout the entire manuscript)